# Towards Robust Multi-Objective Optimization: Adversarial Attack and Defense Methods for Neural Solvers

## Abstract

Deep reinforcement learning (DRL) has shown great promise in addressing multi-objective combinatorial optimization problems. Nevertheless, the robustness of DRL-based neural solvers remain insufficiently explored, especially across diverse and complex problem distributions. This work provides a novel preference-based adversarial attack method, which aims to generate hard problem instances that expose vulnerabilities of solvers. We measure the vulnerability of a solver by evaluating the extent to which its performance in terms of hypervolume deteriorates when tested on hard instances. To mitigate the adversarial effect, we propose a defense method that integrates hardness-aware preference selection into training, leading to substantial improvements in solver robustness and generalizability. The experimental results on multi-objective traveling salesman problem (MOTSP), multi-objective capacitated vehicle routing problem (MOCVRP), and multi-objective knapsack problem (MOKP) verify that our attack method successfully learns hard instances for different solvers. Furthermore, our defense method significantly strengthens the robustness and generalizability of neural solvers, delivering superior performance on hard or out-of-distribution instances.

## 1 Introduction

Deep reinforcement learning (DRL) has emerged as a transformative approach to address combinatorial optimization problems (COPs), which has attracted significant attention in recent years. Its distinct advantages include exceptional computational efficiency, the ability to exploit intrinsic problem structures, and adaptability through iterative feedback-driven learning. Unlike supervised learning Zoph & Le (2017), DRL eliminates the dependence on labeled datasets. Moreover, compared to heuristic and exact algorithms Xin et al. (2021), DRL demonstrates superior efficiency in identifying near-optimal solutions in a reasonable computational time. These attributes make DRL particularly effective in solving classical NP-hard COPs Mazyavkina et al. (2021) Kool et al. (2019) Kwon et al. (2020) Li et al. (2021b), such as the traveling salesman problem (TSP), the knapsack problem (KP), and the capacitated vehicle routing problem (CVRP), as well as their respective multi-objective variants, known as multi-objective combinatorial optimization problems (MOCOPs).

Despite promising performance, current DRL models for COPs often experience significant degradation when faced with instances that differ in distribution or size from their training instances, highlighting their vulnerability to such variations. Hence, the robustness is a major concern to be addressed in developing DRL models for enhancing out-of-distribution generalizability. For single-objective COPs, previous studies Zhang et al. (2022) Lu et al. (2023) found that the performance of DRL models tends to degrade under non-i.i.d. conditions. They mitigated the vulnerability using different strategies, such as altering training strategies and refining model architectures.

For MOCOPs, current DRL models inherently suffer from the same robustness issue (see empirical evidence in Section 3.3). When distributions of training and testing instances diverge, DRL models are prone to overfitting to characteristics of the training distribution, limiting their generalizability on out-of-distribution instances. However, the robustness of DRL models for MOCOPs has not been studied, warranting further research to better understand their vulnerability to different distributions and propose techniques for enhancing their robustness.

In this paper, we first present a preliminary study on the robustness of DRL models for MOCOPs. Our findings demonstrate the vulnerability of DRL models to distributional shifts by revealing a significant impact of the underlying distribution of instances on the hypervolume (HV). This motivates the need for methodologies to robustify DRL models for MOCOPs. Our main contributions can be summarized as follows::

- We introduce a Preference-based Adversarial Attack (PAA) method to target DRL models for MOCOPs. PAA undermines DRL models by generating hard instances that degrade solutions of subproblems associated with specific preferences. The generated instances effectively lower the quality of the Pareto fronts in terms of hypervolume.

- We propose a Dynamic Preference-augmented Defense (DPD) method to mitigate the impact of adversarial attacks. By integrating a hardness-aware preference selection strategy into adversarial training, DPD effectively alleviates the overfitting to restricted preference spaces. It enhances the robustness of DRL models, thereby promoting their generalizability across diverse distributions.

- We evaluate our methods on three classical MOCOPs: MOTSP, MOCVRP and MOKP. The PAA method substantially impairs state-of-the-art DRL models, while the DPD method enhances their robustness, resulting in strong out-of-distribution generalizability.

## 2 RELATED WORK

### 2.1 MOCOP SOLVERS

Algorithms to solve MOCOPs (or MOCOP solvers) are typically classified into exact, heuristic, and learning-based methods. Exact algorithms provide Pareto-optimal solutions, but become computationally intractable for large-scale problems Florios & Mavrotas (2014) Halffmann et al. (2022). Heuristic methods, particularly evolutionary algorithms Zhang & Li (2007b) Fang et al. (2020) Ke et al. (2014) Seada & Deb (2015), effectively explore the solution space through crossover and mutation operations Deb & Jain (2013) Tian et al. (2021), generating a finite set of approximate Pareto solutions in acceptable time. However, their reliance on problem-specific, hand-crafted designs limits their applicability Zhang & Li (2007a).

Learning-based solvers, particularly those based on deep reinforcement learning, have seen growing adoption in MOCOPs, largely due to their high performance and efficiency. Current research on DRL-based neural solvers belongs mainly to two paradigms: one-to-one and many-to-one. In the one-to-one paradigm, each subproblem is addressed by an individual neural solver Wu et al. (2020) Li et al. (2021a) Zhang et al. (2021). In contrast, the many-to-one paradigm streamlines the computational process by using a shared neural solver to handle multiple subproblems Lin et al. (2022) Chen et al. (2024) Fan et al. (2024) Wu et al. (2024) Chen et al. (2025), which outperforms the one-to-one paradigm and delivers state-of-the-art neural solvers. Among them, the efficient meta neural heuristic (EMNH) Chen et al. (2024) learns a meta-model that is rapidly adapted to each preference to solve its subproblem. The preference-conditioned multi-objective combinatorial optimization (PMOCO) Lin et al. (2022) uses a hypernetwork to generate decoder parameters tailored to each subproblem. The conditional neural heuristic (CNH) Fan et al. (2024) leverages dual attention, while the weight embedding model with conditional attention (WE-CA) Chen et al. (2025) employs feature-wise affine transformations, to enhance preference–instance interaction within the encoder. Our study demonstrates that the proposed attack and defense framework is sufficiently general to challenge and robustify models from all three categories.

### 2.2 ROBUSTNESS OF DRL MODELS FOR COPS

Robustness COPs have been studied from both theoretical and neural perspectives. From the theoretical side, Varma & Yoshida (2021) introduced the notion of average sensitivity, measuring the stability of algorithmic outputs under edge deletions in classical COPs such as minimum cut and maximum matching. On the neural side, several studies have investigated hard instance generation and defense methods to improve the robustness of DRL solvers for COPs. For example, Geisler et al. (2021) proposed an efficient and sound perturbation model that adversarially inserts nodes into TSP instances to maximize the deviation between the predicted route and the optimal solution.

Zhang et al. (2022) developed hardness-adaptive curriculum learning methods (HAC) to assess the hardness of given instances and then generate hard instances during training based on the relative difficulty of the solver. Lu et al. (2023) introduced a no-worse optimal cost guarantee (i.e., by lowering the cost of a partial problem) and generated adversarial instances through edge modifications in the graph. In contrast to these approaches that focused on generating hard instances, Zhou et al. (2024) focused on defending neural COP solvers, by an ensemble-based collaborative neural framework designed to improve performance simultaneously in both clean and hard instances.

## 3 PRELIMINARIES

### 3.1 PROBLEM STATEMENT

The mathematical formulation of an MOCOP is generally given as:

$$\min_{\pi \in \mathcal{X}} F(\pi) = \big(f_1(\pi), f_2(\pi), \ldots, f_m(\pi)\big), \tag{1}$$

where $\mathcal{X}$ represents the set of all feasible solutions, and $F(\pi)$ is an $m$-dimensional vector of objective values. A solution $\pi$ to an MOCOP is considered feasible if and only if it satisfies all the constraints specified in the problem. For example, the MOTSP is defined on a graph $G$ with a set of nodes $V = \{0, 1, 2, \ldots, n\}$. Each solution $\pi = (\pi_1, \pi_2, \ldots, \pi_T)$ is a tour consisting of a sequence of nodes of length $T$, where $\pi_j \in V$. The definitions of MOTSP, MOCVRP and MOKP are given in Appendix B.

For multi-objective optimization problems, the goal is to find pareto-optimal solutions that simultaneously optimize all objectives. These pareto-optimal solutions aim to balance trade-offs under different preferences for the objectives. In this paper, we use the following the Pareto concepts Qian et al. (2013):

***Definition 1 (Pareto Dominance).*** Let $u, v \in \mathcal{X}$. The solution $u$ is defined as dominating solution $v$ (denoted $u \prec v$) if and only if, for every objective $i$ where $i \in \{1, \ldots, m\}$, the objective value $f_i(u)$ is less than or equal to $f_i(v)$, and there exists at least one objective $j$ where $j \in \{1, \ldots, m\}$, such that $f_j(u) < f_j(v)$.

***Definition 2 (Pareto Optimality).*** A solution $x^* \in \mathcal{X}$ is Pareto optimal if it is not dominated by any other solution in $\mathcal{X}$. Formally, there exists no solution $x' \in \mathcal{X}$ such that $x' \prec x^*$. The set of all Pareto-optimal solutions is referred to as the Pareto set $\mathcal{P} = \{x^* \in \mathcal{X} \mid \nexists x' \in \mathcal{X} \text{ such that } x' \prec x^*\}$. The projections of Pareto set into the objective space constitute Pareto front $\mathcal{PF} = \{F(x) \mid x \in \mathcal{P}\}$.

### 3.2 DECOMPOSITION

By scalarizing a multi-objective COP into a series of single-objective problems under different preferences, decomposition is an effective strategy for obtaining the Pareto front in DRL models for MOCOPs. Given a preference vector $\lambda = (\lambda_1, \lambda_2, \ldots, \lambda_m) \in \mathbb{R}^m$ with $\lambda_i \geq 0$ and $\sum_{i=1}^{m} \lambda_i = 1$, the weighted sum (WS) and Tchebycheff decomposition (TCH) can be used to transform an MOCOP into scalarized subproblems, which are solved to approximate the Pareto front.

**WS Decomposition.** WS decomposition minimizes convex combinations of $m$ objective functions under preference vectors, as defined below:

$$g_w(\pi|\lambda) = \sum_{i=1}^{m} \lambda_i f_i(\pi), \quad \text{with } \pi \in \mathcal{X}. \tag{1}$$

**Tchebycheff Decomposition.** Tchebycheff decomposition minimizes the maximum weighted distance between the objective values and an ideal point, defined as:

$$g_t(\pi|\lambda, z^*) = \max_{1 \leq i \leq m} \lambda_i |f_i(\pi) - z_i^*|, \quad \text{with } \pi \in \mathcal{X}, \tag{2}$$

where $z^* = (z_1^*, z_2^*, \ldots, z_m^*)$ represents the ideal point, with $z_i^* = \min_{\pi \in \mathcal{X}} f_i(\pi)$.

The decomposition strategy addresses an MOCOP by reducing it to a series of subproblems under varying preferences. Given an instance $x$ and a preference $\lambda$, neural MOCOP solvers learn a stochastic policy $p_\theta$ to approximate the Pareto solution $\pi = (\pi_1, \pi_2, \ldots, \pi_T)$, where $\theta$ represents the learnable parameters.

### 3.3 GLIMPSE OF ROBUSTNESS OF DRL SOLVERS

Given the unexplored robustness of DRL models for MOCOPs, we first examined the performance of two representative neural solvers, PMOCO Lin et al. (2022) and CNH Fan et al. (2024), for MOTSP. Both solvers are pretrained on clean 50-node bi-/tri-objective TSP instances (as in the original papers).

Concretely, we create out-of-distribution test instances using a Gaussian-mixture generator (see details in Appendix A) for evaluation. We vary $c_{\text{DIST}} \in \{1, 5, 10, 20, 30, 40, 50\}$, where $c_{\text{DIST}}$ controls the spatial spread of the clusters, which determines the hardness of the instances. WS-LKH Tinós et al. (2018) (the state-of-the-art solver for MOCOPs) was used as a baseline for comparison. Figure 1 illustrates the HV gaps, representing the difference between the solutions produced by a neural solver and those using WS-LKH:

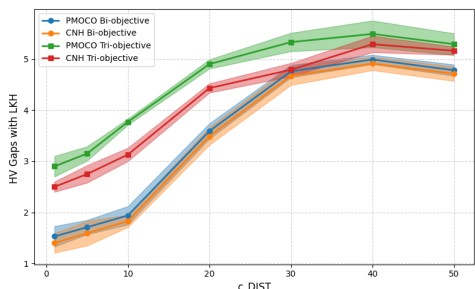

Figure 1: Results for Varying $c_{\text{DIST}}$ in Gaussian Mixture Generator.

$$\text{Gap} = \frac{HV_{\text{LKH}} - HV_{\text{DRL}}}{HV_{\text{LKH}}} \times 100. \tag{3}$$

Our findings reveal that with increasing $c_{\text{DIST}}$ (indicating more difficult test instances), the performance of the neural solvers deteriorates and the gap between their solutions and those provided by WS-LKH widens. These results highlight a significant limitation that neural solvers trained on uniformly distributed instances struggle to maintain robustness as test instances become more diverse and complex.

## 4 THE METHOD

In this section, we introduce a preference-based adversarial attack method to generate hard instances to reflect the robustness of neural solvers. Furthermore, we propose a dynamic preference-augmented defense method to robustify neual solvers. The sketch of the proposed adversarial attack and defense methods is illustrated in Figure 2.

### 4.1 PREFERENCE-BASED ADVERSARIAL ATTACK (PAA)

Typically, neural solvers decompose an MOCOP into a series of subproblems under different preferences, which are solved independently. According to Lin et al. (2022), if a neural solver can solve subproblems (2) well with any preference $\lambda$, it can generate a good approximation to the whole Pareto front for MOCOP. In this paper, we hypothesize that if a neural model does not effectively approximate the solution of the subproblem under certain values of $\lambda$, the resulting approximation of the Pareto front will be inadequate. Following this inspiration, we propose the PAA method to attack neural solvers for MOCOPs. In particular, perturbations are tailored to the original data (i.e., clean instances) in accordance with respective preferences, resulting in hard instances aligned with each specified preference. After identifying hard instances across varying preferences, we gather them into a comprehensive set, which is used to systematically undermine the robustness of a neural solver.

Without loss of generality, we evaluate the performance of a neural solver for each subproblem by generating hard instances under the corresponding preference, which maximize a variant of the reinforcement loss, defined as:

$$\ell(x; \theta) = \frac{L(\pi \mid x)}{b(x)} \log p_\theta(\pi \mid x), \tag{4}$$

where $L(\pi \mid x)$ represents the loss of the subproblem with a given preference $\lambda$ (e.g., using the Tchebycheff decomposition as in Eq. (2)). $b(x)$ is the baseline of $L(\pi \mid x)$, which is calculated by $b(x) = \frac{1}{M} \sum_{j=1}^{M} L(\pi_j \mid x)$, where $M$ is the number of sampled tours for a batch. $x$ denotes the

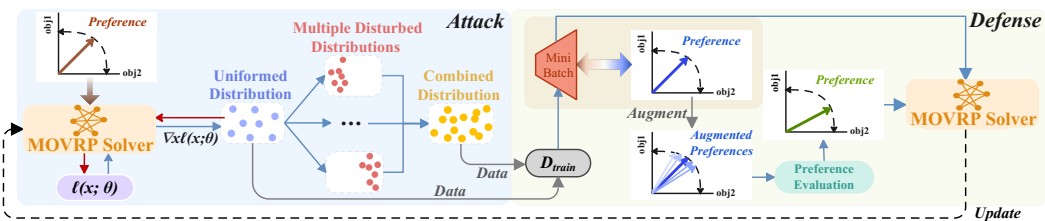

Figure 2: Attack and Defense of Neural Solvers for MOCOP.

problem-specific input, i.e., node coordinates in TSP. $p_\theta(\pi \mid x)$ denotes the probability distribution of the solution $\pi$, which is derived from a neural model parameterized by $\theta$.

Subsequently, the input $x$ corresponding to each preference undergoes the following iterative update:

$$x^{(t+1)} = \Pi_{\mathcal{N}} \left[ x^{(t)} + \alpha \cdot \nabla_{x^{(t)}} \ell(x^{(t)}; \theta^{(t)}) \right], \tag{5}$$

where $\theta^{(t)}$ denotes the best-performing model at iteration $t$, $\mathcal{N}$ represents the feasible solution space, $\alpha$ is the step size, and $\ell(x^{(t)}; \theta^{(t)})$ is the reinforcement loss defined in Eq.(4). At each iteration $t$, the variable $x^{(t)}$ is updated by performing gradient ascent on the loss function $\ell(x; \theta)$, with the calculated gradients $\nabla_{x^{(t)}} \ell(x^{(t)}; \theta^{(t)})$ guiding the update step. The projection operator $\Pi_{\mathcal{N}}(\cdot)$ is a min-max normalization, ensuring the updated variables $x^{(t+1)}$ remains within a feasible solution space $\mathcal{N}$. The iterative process continues until the variable $x^{(t)}$ converges towards hard instances for the current given preference.

In summary, the clean instances $x$ are initially sampled from a uniform distribution, i.e., the distribution of training instances used by neural solvers. Subsequently, we perturb clean instances using our PAA method under respective preferences $\lambda$ to generate diverse hard instances. Ultimately, we gather these instances to construct the set of hard instances $\mathcal{D}_{\text{hard}}$, which are used to assess the robustness of the model.

## 4.2 DYNAMIC PREFERENCE-AUGMENTED DEFENSE (DPD)

To enhance the robustness of the model, we propose the DPD method, which leverages hard instances $\mathcal{D}_{\text{hard}}$ and employs the hardness-aware preference selection method to train the MOCOP solvers.

**Perturbative Preference Generation.** During the adversarial training phase, for each batch in an epoch, we sample a subset of instances $\mathcal{D}_{\text{hard}}^\lambda$ from $\mathcal{D}_{\text{hard}}$ along with the corresponding preferences $\lambda$. Given a preference vector $\lambda = (\lambda_1, \lambda_2, \ldots, \lambda_m)$, we generate a set of augmented preferences $\{\lambda'_1, \lambda'_2, \ldots, \lambda'_m\}$ to explore the neighborhood of $\lambda$. These augmented preferences are dynamically adjusted to emphasize regions where the model exhibits weaker performance. For each preference vector $\lambda$, its augmented preferences are computed as:

$$\lambda'_i = \text{Perturb}(\lambda, \delta_i), \tag{6}$$

where $\delta_i \sim \text{Uniform}(-\epsilon, \epsilon)$ is a small random perturbation. $i$ reflects the index of the perturbed preference vector.

Since the perturbation may result in a preference vector that does not satisfy the constraint $\sum_{k=1}^m \lambda'_{i,k} = 1$, a normalization step is applied to ensure validity:

$$\lambda'_{i,k} = \frac{\lambda'^{\text{raw}}_{i,k}}{\sum_{j=1}^m \lambda'^{\text{raw}}_{i,j}}, \quad \forall k \in \{1, \ldots, m\}, \tag{7}$$

where $\lambda'^{\text{raw}}_{i,k}$ represents the raw preference value after perturbation. By incorporating the normalization step, the generated preferences remain within the valid preference space, ensuring $\sum_{k=1}^m \lambda'_{i,k} = 1$ for all augmented preferences.

**Hardness-aware Preference Selection.** For each augmented preference $\lambda'_i$ and hard instance $x \in D_{\text{hard}}^\lambda$, the neural solver makes an inference to derive a specific Tchebycheff value (Tch). $Tch(\lambda'_i)$

---

**Algorithm 1 Adversarial Training Framework**

---

**Input:** pre-trained model $\theta$, preference set $\Lambda = \{\lambda_k\}_{k=1}^P$, hard instances $\mathcal{D}_{\text{hard}} = \emptyset$, epochs $E$, batch size $B$, number of perturbed preferences $N$, optimizer ADAM, size of mini-batches per epoch $M$

**Output:** Updated model parameters $\theta$

1:  **for** $t = 1$ to $E$ **do**                                                                              $\triangleright$ Epoch loop
2:      Generate $\mathcal{D}_{\text{clean}}$ with uniform distribution.
3:      **for** $k = 1$ to $P$ **do**
4:          Select preference $\lambda_k \in \Lambda$.
5:          Generate $d^{\text{adv},k}$ for $\mathcal{D}_{\text{clean}}$ using preference $\lambda_k$ via PAA.
6:          $\mathcal{D}_{\text{hard}} = \mathcal{D}_{\text{hard}} \cup d^{\text{adv},k}$.
7:      **end for**
8:      $\mathcal{D}_{\text{train}} = \mathcal{D}_{\text{hard}} \cup \mathcal{D}_{\text{clean}}$.
9:      **for** $j = 1$ to $M$ **do**                                                                         $\triangleright$ Mini-batch loop
10:         Sample mini-batch $\mathcal{B} \subset \mathcal{D}_{\text{train}}$ of size $B$, each with preference $\lambda_j$.
11:         **for** $i = 1$ to $N$ **do**                                                                     $\triangleright$ Preference augmentation
12:             Generate $\lambda_i'$ using (6) and (7) and estimate Tch values for $\lambda_i'$ using $\mathcal{B}$.
13:         **end for**
14:         Select $\lambda_{\text{adv}}$ according to Eqs. (8) and (9).
15:         Compute gradient $\nabla \mathcal{J}(\theta)$ using $\lambda_{\text{adv}}$ and $\mathcal{B}$ (Eq. (11)).
16:         Update parameters: $\theta \leftarrow \text{ADAM}(\theta, \nabla \mathcal{J}(\theta))$.
17:     **end for**
18: **end for**

---

quantifies the quality of the solution generated by the model with preference $\lambda_i'$, where a smaller $Tch(\lambda_i')$ indicates a better quality solution for this preference. $Tch(\lambda_i')$ are processed using the following softmax function to compute a relevance score for each preference:

$$P(\lambda_i') = \frac{\exp(-Tch(\lambda_i'))}{\sum_{j=1}^N \exp(-Tch(\lambda_j'))}. \tag{8}$$

The preferences $\lambda_i'$ that produce the poorest solutions (i.e., the highest $Tch(\lambda_i')$) are selected for further optimization. $N$ in Eq.(8) denotes the total number of augmented preference vectors.

The preferences associated with the lowest relevance scores, as identified through $P(\lambda_i')$, signify regions that cause lower performance of the solver. These preferences are prioritized for further being involved in the optimization, aiming to improve the robustness and generalizability of the model across diverse preferences. The preference with the smallest $P(\lambda_i')$ is selected for further optimization:

$$\lambda_{\text{adv}}' = \arg\min_i P(\lambda_i'). \tag{9}$$

For the selected adversarial preference $\lambda_{\text{adv}}'$, the original hard instances $D_{\text{hard}}^\lambda$ are reused for training. For each instance $x \in D_{\text{hard}}^\lambda$, the loss function is recalibrated as:

$$L(x \mid \lambda_{\text{adv}}') = \max_{1 \le k \le m} \lambda_{\text{adv},k}' \cdot |f_k(x) - z_k^*|. \tag{10}$$

**Training Framework.** The proposed adversarial training framework is detailed in Algorithm 1. Each epoch in our training framework comprises two phases: the attack phase and the defense phase. During the attack phase, hard instances tailored to individual preferences are generated and subsequently aggregated for further training. In the defense phase, neural MOCOP solvers are trained on constructed instances. The optimization process employs the REINFORCE algorithm Williams (1992) to minimize the loss, which is formulated as follows.

$$\nabla \mathcal{J}(\theta) = \mathbb{E}_{\pi \sim p_\theta,\, s \sim \mathcal{D}_{\text{train}},\, \lambda_{\text{adv}} \sim \mathcal{S}_\lambda} \left[ \left( L(\pi \mid s, \lambda_{\text{adv}}) - L_b(s, \lambda_{\text{adv}}) \right) \cdot \nabla \log p_\theta(\pi \mid s, \lambda_{\text{adv}}) \right] \tag{11}$$

where $L_b(s, \lambda_{\text{adv}})$ is used as a baseline to reduce the variance in the estimation of the gradient. Monte Carlo sampling is used to approximate this expectation, where diverse training samples and randomly selected preferences are used iteratively to optimize the model parameters.

## 5 EXPERIMENTS

In this section, we conduct a comprehensive set of experiments on four MOCOPs (Appendix B): bi-objective TSP (Bi-TSP), tri-objective TSP (Tri-TSP), bi-objective CVRP (Bi-CVRP), and bi-objective KP (Bi-KP) to thoroughly analyze and evaluate the effectiveness of the proposed attack and defense methods. All experiments are executed on a server equipped with an Intel(R) Xeon(R) Silver 4214R CPU @ 2.40GHz and an RTX 3090 GPU.

### 5.1 BASELINES AND SETTINGS

**Instance Distributions for Evaluation.** To evaluate the efficacy of the proposed attack approach, we benchmark against four typical instance distributions (clean uniform, log-normal (0,1) with moderate skewness, beta (2,5) with bounded asymmetry, and gamma (2,0.5) with high skewness) as well as ROCO Lu et al. (2023), a learning-based attack method that perturbs graph edges under a no-worse-optimum guarantee and trains an agent with PPO Schulman et al. (2017) to maximize solver degradation.

**Evaluation Setup for Targeted Solvers.** We target state-of-the-art neural MOCOP solvers, namely Conditional Neural Heuristic **CNH**[1] , Meta Neural Heuristic **EMNH** [2] Chen et al. (2024), and Preference-Based Neural Heuristic **PMOCO**[3]. We selected these solvers as they all adopt **POMO** Kwon et al. (2020) as the base model for solving single-objective subproblems. For fair comparisons, we adopt WS (weighted sum) scalarization across all methods. To establish the baseline for the relative optimality gap, we approximate the Pareto front using two non-learnable solvers: **WS-LKH** for MOTSP and MOCVRP, and weighted-sum dynamic programming (**WS-DP**) for MOKP.

**Metrics.** To evaluate the proposed attack and defense methods, the average **HV** Audet et al. (2021) and the average optimality **gap** are employed. HV provides a comprehensive measure of both the diversity and convergence of solutions, while the gap quantifies the relative difference in HV compared to the first baseline solver.

**Implementations.** We evaluate PMOCO, CNH, and EMNH using their pre-trained models. Hard instances are generated with 101 and 105 uniformly sampled preferences for the bi- and tri-objective settings, respectively, with 100 clean samples per preference, yielding 10,100 and 10,500 instances. Training uses 10,000 clean samples plus hard instances per epoch, with 3 gradient steps (step size 0.01) over 200 epochs. An ablation on these parameters is given in Appendix C. For testing, 200 Gaussian instances are constructed with $c_{\text{DIST}} \in [10, 20, 30, 40, 50]$. Other settings (e.g., learning rate, batch size) follow their original papers.

### 5.2 ATTACK PERFORMANCE

From Table 1, it can be observed that perturbations based on log-normal, beta, and gamma distributions generally have little effect on reducing the HV value of the solution set. In particular, these perturbations produce higher HV values across various solvers compared to clean instances. This indicates that conventional disturbances struggle to substantially impair the performance of solvers such as WS-LKH and WS-DP. Furthermore, the discrepancies between the solutions generated by these neural solvers and the conventional solver under these distributions are consistently smaller than those observed for clean instances. Hence, despite the heterogeneous nature of these distributions, neural solvers demonstrate robust capabilities to maintain high-quality solutions. In contrast, PAA generates problem distributions that significantly reduce HV values in both classical and neural MOCOP solvers, demonstrating strong and consistent attack effect across all problems and sizes. Notably, it achieves the best attack effect over all cases in Bi-KP.

Furthermore, the HV gaps of different solvers on hard instances generated by PAA and on clean instances are considerably larger. For example, on Bi-CVRP50, the attack against EMNH yields a gap of 4.73%, while on Bi-KP100, the attack against PMOCO reaches 7.09%, significantly exceeding the attack effects by instances generated by the other methods.

---

[1] https://github.com/mingfan321/CNH/
[2] https://github.com/bill-cjb/EMNH
[3] https://github.com/Xi-L/PMOCO

Table 1: Optimality Gap Analysis for Attack Performance. **Bold** values indicate the best performance.

| Method | Size | Optimality Gap (%) | | | | | | | | | | | |
| | | Clean instances | | LogNormal(0,1) | | Beta(2, 5) | | Gamma(2, 0.5) | | ROCO-RL | | PAA | |
| | | HV(↓) | Gap(↑) | HV(↓) | Gap(↑) | HV(↓) | Gap(↑) | HV(↓) | Gap(↑) | HV(↓) | Gap(↑) | HV(↓) | Gap(↑) |
| **Bi-TSP** | | | | | | | | | | | | | |
| WS-LKH | 20 | 0.5118 | - | 0.8503 | - | 0.6566 | - | 0.7348 | - | 0.5627 | - | 0.4675 | - |
| | 50 | 0.5757 | - | 0.8760 | - | 0.7012 | - | 0.7768 | - | 0.6231 | - | 0.5608 | - |
| | 100 | 0.6799 | - | 0.9123 | - | 0.7201 | - | 0.7771 | - | 0.7094 | - | 0.6824 | - |
| EMNH | 20 | 0.5042 | 1.48% | 0.8482 | 0.25% | 0.6514 | 0.79% | 0.7301 | 0.64% | 0.5568 | 1.05% | 0.4591 | **1.79%** |
| | 50 | 0.5671 | 1.49% | 0.8731 | 0.33% | 0.6954 | 0.83% | 0.7750 | 0.23% | 0.6141 | 1.44% | 0.5502 | **1.89%** |
| | 100 | 0.6653 | 2.15% | 0.9034 | 0.97% | 0.7119 | 1.14% | 0.7726 | 0.58% | 0.6914 | **2.53%** | 0.6665 | 2.33% |
| PMOCO | 20 | 0.5038 | 1.55% | 0.8477 | 0.31% | 0.6485 | 1.25% | 0.7291 | 0.73% | 0.5535 | 1.64% | 0.4584 | **1.95%** |
| | 50 | 0.5651 | 1.85% | 0.8691 | 0.78% | 0.6951 | 0.88% | 0.7708 | 0.77% | 0.6113 | 1.88% | 0.5498 | **1.96%** |
| | 100 | 0.6571 | **3.34%** | 0.8948 | 1.92% | 0.7035 | 2.33% | 0.7705 | 0.84% | 0.6946 | 2.09% | 0.6603 | 3.23% |
| CNH | 20 | 0.5061 | 1.11% | 0.8499 | 0.04% | 0.6557 | 1.81% | 0.7321 | 0.36% | 0.5536 | **1.62 %** | 0.4603 | 1.54% |
| | 50 | 0.5669 | 1.53% | 0.8722 | 0.43% | 0.6949 | 0.89% | 0.7721 | 0.60% | 0.6124 | 1.72% | 0.5507 | **1.80%** |
| | 100 | 0.6682 | 1.64% | 0.9049 | 0.81% | 0.7130 | 0.98% | 0.7741 | 0.38% | 0.6957 | 1.93% | 0.6621 | **2.12%** |
| **Bi-CVRP** | | | | | | | | | | | | | |
| WS-LKH | 20 | 0.2466 | - | 0.7145 | - | 0.8164 | - | 0.7158 | - | 0.3225 | - | 0.2450 | - |
| | 50 | 0.3140 | - | 0.7255 | - | 0.8361 | - | 0.7285 | - | 0.3690 | - | 0.3402 | - |
| | 100 | 0.2408 | - | 0.7265 | - | 0.7435 | - | 0.7252 | - | 0.3782 | - | 0.2705 | - |
| EMNH | 20 | 0.2404 | 2.51% | 0.6955 | 2.66% | 0.8022 | 1.74% | 0.6958 | 2.79% | 0.3149 | 2.33% | 0.2354 | **3.92%** |
| | 50 | 0.3048 | 2.93% | 0.7179 | 1.05% | 0.8235 | 1.51% | 0.7090 | 2.68% | 0.3609 | 2.18% | 0.3241 | **4.73%** |
| | 100 | 0.2309 | 4.11% | 0.7240 | 0.34% | 0.7363 | 0.97% | 0.7149 | 0.04% | 0.3709 | 1.93% | 0.2588 | **4.32%** |
| PMOCO | 20 | 0.2415 | 2.07% | 0.7090 | 0.77% | 0.8139 | 0.31% | 0.7100 | 0.77% | 0.3133 | **2.84%** | 0.2401 | 2.00% |
| | 50 | 0.3081 | 1.88% | 0.7220 | 0.48% | 0.8330 | 0.37% | 0.7224 | 0.84% | 0.3573 | 3.17% | 0.3286 | **3.41%** |
| | 100 | 0.2307 | 4.19% | 0.7248 | 0.23% | 0.7375 | 0.81% | 0.7050 | 2.77% | 0.3703 | 2.08% | 0.2591 | **4.21%** |
| CNH | 20 | 0.2457 | 0.36% | 0.7149 | -0.06% | 0.8157 | 0.08% | 0.7140 | 0.25% | 0.3168 | 1.77% | 0.2404 | 1.87% |
| | 50 | 0.3090 | 1.59% | 0.7250 | 0.07% | 0.8340 | 0.25% | 0.7291 | -0.08% | 0.3615 | 2.02% | 0.3295 | **3.15%** |
| | 100 | 0.2393 | 0.62% | 0.7235 | 0.41% | 0.7389 | 0.62% | 0.7179 | 1.00% | 0.3718 | 1.69% | 0.2597 | **3.99%** |
| **Bi-KP** | | | | | | | | | | | | | |
| WS-DP | 50 | 0.7122 | - | 0.6809 | - | 0.6851 | - | 0.8138 | - | - | - | 0.6212 | - |
| | 100 | 0.8283 | - | 0.6628 | - | 0.6046 | - | 0.8012 | - | - | - | 0.6483 | - |
| | 200 | 0.6384 | - | 0.5799 | - | 0.5922 | - | 0.7799 | - | - | - | 0.3929 | - |
| EMNH | 50 | 0.7270 | -2.07% | 0.6883 | -1.09% | 0.6821 | 0.44% | 0.8093 | 0.55% | - | - | 0.5918 | **4.73%** |
| | 100 | 0.8571 | -3.47% | 0.6827 | -3.00% | 0.6189 | -2.36% | 0.7984 | 0.35% | - | - | 0.6056 | **6.59%** |
| | 200 | 0.6299 | 1.33% | 0.5803 | -0.01% | 0.6074 | -2.57% | 0.7894 | -1.22% | - | - | 0.3838 | **2.32%** |
| PMOCO | 50 | 0.7251 | -1.81% | 0.6969 | -2.35% | 0.6808 | 0.63% | 0.8231 | -1.14% | - | - | 0.5901 | **5.00%** |
| | 100 | 0.8549 | -3.22% | 0.7058 | -6.49% | 0.6241 | -3.21% | 0.8181 | -2.11% | - | - | 0.6023 | **7.09%** |
| | 200 | 0.6295 | 1.39% | 0.5769 | 0.52% | 0.6019 | -1.64% | 0.7706 | 1.19% | - | - | 0.3816 | **2.88%** |
| CNH | 50 | 0.7313 | -2.68% | 0.6982 | -2.54% | 0.7023 | -2.51% | 0.8251 | -1.39% | - | - | 0.5957 | **4.10%** |
| | 100 | 0.8608 | -3.93% | 0.7028 | -6.04% | 0.6239 | -3.19% | 0.8194 | -2.27% | - | - | 0.6094 | **6.00%** |
| | 200 | 0.6301 | 1.30% | 0.5822 | -0.39% | 0.6127 | -3.46% | 0.7853 | -0.69% | - | - | 0.3857 | **2.24%** |
| **Tri-TSP** | | | | | | | | | | | | | |
| WS-LKH | 20 | 0.3279 | - | 0.7841 | - | 0.5065 | - | 0.6156 | - | 0.3740 | - | 0.2615 | - |
| - | 50 | 0.3557 | - | 0.8049 | - | 0.5462 | - | 0.6647 | - | 0.4229 | - | 0.3161 | - |
| | 100 | 0.4599 | - | 0.8663 | - | 0.6392 | - | 0.7392 | - | 0.4874 | - | 0.4490 | - |
| EMNH | 20 | 0.3218 | 1.86% | 0.7753 | 1.12% | 0.5039 | 0.51% | 0.6119 | 0.60% | 0.3679 | 1.63% | 0.2549 | **2.52%** |
| | 50 | 0.3418 | 3.90% | 0.7942 | 1.33% | 0.5402 | 1.09% | 0.6499 | 2.23% | 0.4070 | 3.74% | 0.3022 | **4.39%** |
| | 100 | 0.4412 | 4.07% | 0.8309 | 4.08% | 0.6114 | 4.35% | 0.7118 | 3.71% | 0.4659 | **4.41%** | 0.4296 | 4.32% |
| PMOCO | 20 | 0.3228 | 1.54% | 0.7799 | 0.52% | 0.5020 | 0.88% | 0.6114 | 0.68% | 0.3675 | 1.73% | 0.2563 | **1.98%** |
| | 50 | 0.3425 | 3.72% | 0.7968 | 1.01% | 0.5337 | 2.28% | 0.6512 | 2.03% | 0.4089 | 3.29% | 0.3031 | **4.13%** |
| | 100 | 0.4349 | 5.42% | 0.8324 | 3.92% | 0.6141 | 3.93% | 0.7112 | 3.78% | 0.4620 | 5.21% | 0.4237 | **5.63%** |
| CNH | 20 | 0.3241 | 1.15% | 0.7804 | 0.47% | 0.5036 | 0.57% | 0.6126 | 0.48% | 0.3681 | **1.58%** | 0.2579 | 1.37% |
| | 50 | 0.3433 | 3.48% | 0.7984 | 0.80% | 0.5408 | 0.98% | 0.6574 | 1.09% | 0.4080 | 3.52% | 0.3042 | **3.76%** |
| | 100 | 0.4401 | 4.31% | 0.8417 | 2.84% | 0.6180 | 3.31% | 0.7164 | 3.08% | 0.4641 | 4.77% | 0.4253 | **5.27%** |

ROCO-RL shows non-trivial attack capability on a few instances (e.g., a notable 4.41% gap against EMNH on Tri-TSP100), yet PAA consistently surpasses it in most MOCO problems, achieving superior attack performance. This indicates that PAA explicitly exposes the vulnerabilities of diverse neural MOCOP solvers, underscoring its effectiveness in challenging solvers across MOCO problems of different sizes and types.

## 5.3 Defense Performance

To evaluate our defense method, we conducted comparative experiments on EMNH, PMOCO, and CNH trained on uniformly distributed clean instances, alongside their DPD variants trained under the proposed framework. We also included WE-CA Chen et al. (2025), a recent neural solver that employs feature-wise affine transformations at the encoder level, as a state-of-the-art baseline for robustness evaluation. All models (with or without DPD) were evaluated in Gaussian instances. The results are reported in Table 2. As shown, DPD-defended solvers (PMOCO-DPD, CNH-DPD, EMNH-DPD, WE-CA-DPD) consistently enhance the performance of neural solvers, achieving overall improvements on all problems. Remarkably, on Bi-TSP20 and Bi-CVRP100, WE-CA-DPD and CNH-DPD achieve the first and second best results, respectively. The improvement is particularly evident on Bi-CVRP100, where WE-CA-DPD improves HV by 2.23% over WS-LKH, the largest gain among all solvers.

In addition, CNH-DPD achieves the best result on Bi-CVRP50. The strong and consistent Bi-CVRP results indicate that models with encoder-level preference-instance interaction mechanism (e.g., CNH, WE-CA) exhibit the most pronounced improvements under DPD. In particular, CNH-DPD and WE-CA-DPD deliver leading performance on Bi-CVRP20/50/100.

Regarding the meta-learning–based solver, EMNH-DPD improves EMNH performance and produces the best result (HV 0.6885, with a runtime of 5.29s) on Tri-TSP20, as well as second-best results on Tri-TSP50 and Tri-TSP100. This demonstrates the versatility of DPD in enhancing solvers

Table 2: Optimality Gap Analysis for Defense Performance. **Bold** values indicate the best performance in the respective metric. Underlined values indicate the second-best performance in the respective metric.

| Method | Instance | 20 Nodes | | | 50 Nodes | | | 100 Nodes | | |
|---|---|---|---|---|---|---|---|---|---|---|
| | | HV ($\uparrow$) | Gap ($\downarrow$) | Time ($\downarrow$) | HV ($\uparrow$) | Gap ($\downarrow$) | Time($\downarrow$) | HV ($\uparrow$) | Gap ($\downarrow$) | Time ($\downarrow$) |
| WS-LKH | | 0.8873 | - | 4.30m | 0.8660 | - | 38.47m | 0.8365 | - | 3.19h |
| EMNH | | 0.8742 | 1.48% | 6.12s | 0.8649 | 0.13% | 9.42s | 0.8265 | 1.19% | 30.09s |
| **EMNH-DPD** | | **0.8894** | -0.23% | 6.38s | **0.8697** | **-0.43%** | 9.15s | 0.8317 | 0.57% | 30.25s |
| PMOCO | | 0.8779 | 1.06% | 7.11s | 0.8566 | 1.08% | 11.33s | 0.8248 | 1.40% | 30.87s |
| **PMOCO-DPD** | Bi-TSP | 0.8867 | 0.07% | 8.44s | 0.8654 | 0.06% | 13.29s | 0.8360 | 0.06% | 32.08s |
| CNH | | 0.8794 | 0.89% | 7.32s | 0.8587 | 0.84% | 12.03s | 0.8294 | 0.85% | 34.76s |
| **CNH-DPD** | | 0.8871 | 0.02% | 8.67s | 0.8660 | 0.00% | 15.22s | 0.8373 | -0.09% | 33.57s |
| WE-CA | | 0.8803 | 0.78% | 7.33s | 0.8591 | 0.79% | 10.52s | 0.8304 | 0.72% | 31.34s |
| **WE-CA-DPD** | | **0.8886** | **-0.14%** | 7.29s | 0.8674 | -0.16% | 10.17s | **0.8377** | **-0.14%** | 31.41s |
| WS-LKH | | 0.5743 | - | 6.44m | 0.5314 | - | 44.82m | 0.5157 | - | 4.03h |
| EMNH | | 0.5558 | 3.22% | 6.03s | 0.5278 | 0.67% | 16.11s | 0.5048 | 2.11% | 40.29s |
| **EMNH-DPD** | | **0.5772** | -0.50% | 6.72s | 0.5319 | -0.09% | 16.88s | 0.5258 | -1.96% | 40.75s |
| PMOCO | | 0.5526 | 3.78% | 6.39s | 0.5219 | 1.79% | 18.02s | 0.5015 | 2.75% | 47.21s |
| **PMOCO-DPD** | Bi-CVRP | 0.5763 | -0.35% | 6.51s | 0.5308 | 0.11% | 17.44s | 0.5173 | -0.31% | 47.93s |
| CNH | | 0.5564 | 3.11% | 7.23s | 0.5289 | 0.47% | 19.55s | 0.5071 | 1.67% | 52.16s |
| **CNH-DPD** | | 0.5794 | -0.88% | 7.65s | **0.5386** | **-1.35%** | 19.98s | 0.5261 | -2.01% | 51.49s |
| WE-CA | | 0.5572 | 2.97% | 6.41s | 0.5292 | 0.41% | 16.43s | 0.5109 | 0.93% | 44.24s |
| **WE-CA-DPD** | | **0.5803** | **-1.04%** | 6.23s | 0.5382 | -1.27% | 16.37s | **0.5272** | **-2.23%** | 43.72s |
| WS-DP | | **0.5832** | - | 17.45m | **0.4948** | - | 1.42h | **0.6783** | - | 4.23h |
| EMNH | | 0.5817 | 0.26% | 5.32s | 0.4858 | 1.81% | 17.46s | 0.6682 | 1.49% | 40.23s |
| **EMNH-DPD** | | 0.5828 | 0.06% | 5.38s | 0.4903 | 0.91% | 18.45s | 0.6718 | 0.96% | 40.74s |
| PMOCO | | 0.5809 | 0.39% | 7.22s | 0.4803 | 2.93% | 16.87s | 0.6653 | 1.91% | 48.31s |
| **PMOCO-DPD** | Bi-KP | 0.5829 | 0.05% | 7.88s | 0.4897 | 1.03% | 17.94s | 0.6712 | 1.04% | 48.92s |
| CNH | | 0.5820 | 0.21% | 8.17s | 0.4845 | 2.08% | 19.12s | 0.6683 | 1.47% | 54.11s |
| **CNH-DPD** | | **0.5832** | **0.00%** | 8.49s | 0.4901 | 0.95% | 19.46s | 0.6742 | 0.60% | 53.74s |
| WE-CA | | 0.5823 | 0.15% | 7.24s | 0.4852 | 1.94% | 16.18s | 0.6691 | 1.35% | 45.72s |
| **WE-CA-DPD** | | 0.5829 | 0.05% | 7.53s | 0.4913 | 0.70% | 17.44s | 0.6752 | 0.46% | 44.43s |
| WS-LKH | | 0.6864 | - | 6.03m | **0.6151** | - | 55.14m | **0.4978** | - | 3.71h |
| EMNH | | 0.6719 | 2.11% | 5.12s | 0.5831 | 5.20% | 9.43s | 0.4829 | 2.99% | 30.31s |
| **EMNH-DPD** | | **0.6885** | **-0.31%** | 5.29s | 0.6144 | 0.11% | 9.67s | 0.4960 | 0.36% | 30.52s |
| PMOCO | | 0.6708 | 2.27% | 6.18s | 0.5954 | 3.20% | 11.22s | 0.4825 | 3.07% | 30.32s |
| **PMOCO-DPD** | Tri-TSP | 0.6817 | 0.68% | 7.08s | 0.6079 | 1.17% | 12.34s | 0.4930 | 0.96% | 32.21s |
| CNH | | 0.6791 | 1.06% | 7.32s | 0.6049 | 1.65% | 12.67s | 0.4872 | 2.13% | 33.86s |
| **CNH-DPD** | | 0.6877 | -0.18% | 8.11s | 0.6127 | 0.39% | 15.42s | 0.4938 | 0.80% | 33.14s |
| WE-CA | | 0.6793 | 1.03% | 6.33s | 0.6051 | 1.63% | 12.45s | 0.4876 | 2.04% | 31.41s |
| **WE-CA-DPD** | | 0.6862 | 0.03% | 7.43s | 0.6133 | 0.29% | 13.09s | 0.4952 | 0.52% | 31.49s |

across distinct learning paradigms. We further tested DPD to defend neural solvers against the hard instances generated by ROCO-RL, which also exhibited evident robustness improvement (see Appendix D for details).

In terms of computational efficiency, DPD-defended solvers require considerably less runtime compared to non-learnable solvers. For example, WS-DP requires 17.45 minutes to reach the best HV value on Bi-KP, while CNH-DPD in only 8.49 seconds achieves the same. To further validate the robustness of our framework, we evaluated DPD on six Bi-TSP benchmark instances from TSPLIB Reinelt (1991), as well as two large-scale test sets ($n = 150/200$) (see Appendices E and F). Overall, these results demonstrate that DPD substantially enhances the robustness of neural solvers, yielding strong generalization to larger problem sizes and distribution shifts.

## 6 CONCLUSIONS

In this paper, we investigate the robustness and performance of state-of-the-art neural MOCOP solvers under diverse hard and clean instances distributions. We proposed an innovative attack method that effectively generates hard (challenging) problem instances, measuring the vulnerability in solver's performance by reducing HV values and increasing optimality gaps compared to baseline methods. Furthermore, we also proposed a defense method leverages adversarial training with hardness-aware preference selection, showing improved robustness across various solvers and tasks. These two methods contribute to solving multi-objective optimization challenges by enhancing the robustness and generalizability of neural solvers, leading to more robust solutions. In the future, we aim to extend our method to address dynamic real-world MOCOP instances, integrating domain-specific constraints, and improving generalizability in online environments.

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

# A  GAUSSIAN MIXTURE GENERATOR

Instead of uniformly distributing the nodes, the Gaussian mixture generator partitions them into clusters, enabling the creation of TSP instances with varying levels of difficulty. The process begins by determining the number of clusters, $n_c$, sampled from a discrete uniform distribution $U(c_{\min}, c_{\max})$, where $c_{\min}$ and $c_{\max}$ denote the minimum and maximum number of clusters, respectively. Following Zhang et al. (2022), we set $c_{\min} = 3$ and $c_{\max} = 7$ in our experiments. Each node is assigned to one of the $n_c$ clusters with equal probability, ensuring a balanced distribution. The center of each cluster is represented as $\mu_i = (\mu_{i1}, \mu_{i2})$, where $\mu_{i1}$ and $\mu_{i2}$ denote the x- and y-coordinates of the cluster center, respectively. These coordinates are uniformly sampled as:

$$\mu_i \sim U([0, c_{\mathrm{DIST}}]^2), \tag{12}$$

where $c_{\mathrm{DIST}}$ controls the spread of the clusters.

The coordinates of each node, $x_i$, are drawn from a Gaussian distribution $N(\mu_{c_i}, I)$, where $\mu_{c_i}$ represents the center of the cluster $c_i$ to which node $i$ belongs and $I$ is the identity covariance matrix. This ensures that nodes within the same cluster are spatially close to their cluster center. To standardize the coordinates, we apply min-max normalization to scale all nodes into a unit square $[0, 1]^2$:

$$\tilde{x}_i = \frac{x_i - \min(X)}{\max(X) - \min(X)}, \tag{13}$$

where $\min(X)$ and $\max(X)$ are computed dimension-wise across the entire set of nodes $X$. This normalization ensures consistency across instances.

By introducing cluster-based distributions, the Gaussian mixture generator generates TSP instances with diverse spatial structures and controlled levels of complexity, offering a more realistic evaluation of algorithmic robustness compared to uniform sampling.

# B  MULTI-OBJECTIVE COMBINATORIAL OPTIMIZATION PROBLEMS

Multi-Objective combinatorial optimization problems (MOCOPs) extend classical optimization problems by incorporating multiple objectives. This section explores three key problems: the Multi-Objective Traveling Salesman Problem (MOTSP), the Multi-Objective Capacitated Vehicle Routing Problem (MOCVRP), and the Multi-Objective Knapsack Problem (MOKP), each involving the optimization of competing objectives under specific constraints.

## B.1  MULTI-OBJECTIVE TRAVELING SALESMAN PROBLEM (MOTSP)

MOTSP is an extension of the classic single-objective Traveling Salesman Problem (TSP). In MOTSP, $M$ objectives are considered, with each objective represented by a distinct set of node coordinates. The aim is to find a tour $\pi$, which is a cyclic permutation of the nodes, that simultaneously minimizes the costs across all objectives:

$$\min L(\pi|s) = \min(L_1(\pi|s), L_2(\pi|s), \ldots, L_M(\pi|s)), \tag{14}$$

where $L_i(\pi|s)$ denotes the cost for the $i$-th objective and is calculated as:

$$L_i(\pi|s) = c_i(\pi(n), \pi(1)) + \sum_{j=1}^{n-1} c_i(\pi(j), \pi(j+1)). \tag{15}$$

Here, $c_i(j, k)$ represents the cost of moving from node $j$ to node $k$ under the $i$-th objective. The solution to MOTSP often involves trade-offs as it requires minimizing all objective functions simultaneously.

## B.2 MULTI-OBJECTIVE VEHICLE ROUTING PROBLEM (MOCVRP)

MOCVRP aims to optimize two objectives simultaneously: minimizing the total length of the route, which is the sum of distances traveled by all vehicles, and minimizing the makespan, defined as the length of the longest route. This problem involves a depot node and multiple customer nodes, each with a specific demand $q_i$. A fleet of vehicles, each with a fixed capacity $D$, starts and ends its routes at the depot, ensuring that the total demand on any route satisfies the constraint $\sum q_i \leq D$.

The total route length can be mathematically formulated as

$$\min f_1(\pi) = \sum_{k=1}^{K} \sum_{i=1}^{n_k} d_{\pi_k(i), \pi_k(i+1)}, \tag{16}$$

where $K$ denotes the number of vehicles, $n_k$ is the number of customer nodes in the $k$-th route, and $d_{\pi_k(i), \pi_k(i+1)}$ is the distance between consecutive nodes in the route. The makespan, representing the longest route among all vehicles, is expressed as

$$\min f_2(\pi) = \max_{k \in \{1, \dots, K\}} \sum_{i=1}^{n_k} d_{\pi_k(i), \pi_k(i+1)}. \tag{17}$$

In addition, the solution must satisfy two key constraints. Each customer must be visited exactly once, and all routes must start and end at the depot. This problem models real-world scenarios where optimizing operational efficiency and resource utilization is critical in multi-vehicle delivery systems.

## B.3 MULTI-OBJECTIVE KNAPSACK PROBLEM (MOKP)

The Knapsack Problem (KP) is a classic problem in combinatorial optimization, and MOKP is an extension of KP, involving $m$ objectives and $n$ items. The goal of this problem is to maximize the values of multiple objective functions:

$$f(x) = \max(f_1(x), f_2(x), \dots, f_m(x)), \tag{18}$$

where each objective function is defined as

$$f_i(x) = \sum_{j=1}^{n} v_{ij} x_j. \tag{19}$$

The constraints are given by

$$\sum_{j=1}^{n} w_j x_j \leq W, \quad \text{with } x_j \in \{0, 1\}. \tag{20}$$

Each item has a weight $w_j$ and $m$ different values $v_{ij}$, where $i = 1, 2, ..., m$. The knapsack has a maximum weight capacity $W$, and the objective is to select a set of items such that their total weight does not exceed the capacity $W$, while maximizing the sum of values for each objective.

# C ABLATION STUDY

Ablation studies were conducted on critical hyperparameters of the proposed attack method, with experiments performed on three-objective 50-node TSP instances.

## C.1 IMPACT OF GRADIENT ITERATION COUNTS

The iteration count $t$ in Equation (5) of the mian paper is varied from 1 to 10 to evaluate its impact on the HV values and the gap relative to the LKH. As illustrated in Figure 4, the gap peaks at $t = 3$ and $t = 8$, with the maximum observed at $t = 3$. Consequently, $t = 3$ is selected in our experiment to balance computational efficiency and performance analysis.

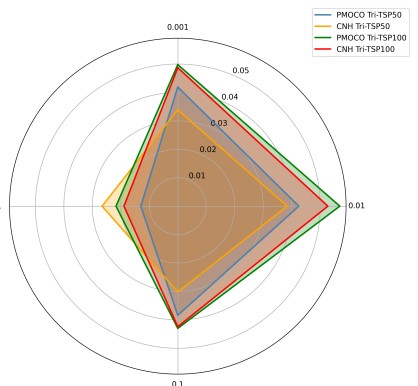

Figure 3: HV Gaps for Different $\alpha$.

## C.2 IMPACT OF GRADIENT UPDATE PARAMETERS

The radar graph 3 illustrates the relationship between the step size $\alpha$ in Equation (5) of the main paper and their HV gaps, showing that the gap reaches its maximum values in $\alpha = 0.01$. Therefore, to maximize the effectiveness of the attack, $\alpha = 0.01$ is adopted in our experiments.

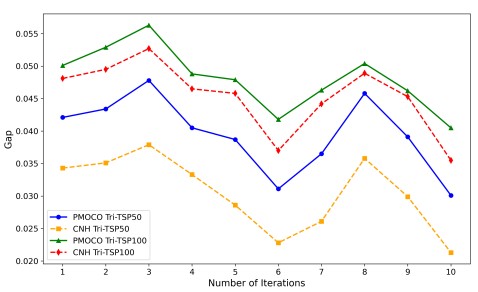

Figure 4: Impact of Iteration Counts on HV and Gap.

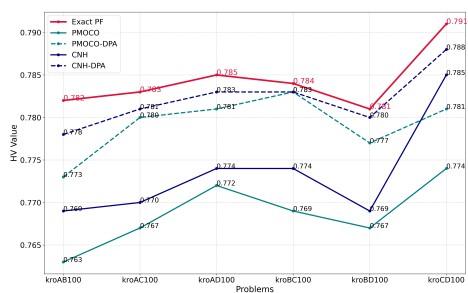

Figure 5: Benchmark Performance Comparison on HV Metric.

## D ROBUST TRAINING ON ROCO-ADVERSARIAL INSTANCES

**Setup.** We adopt an *offline* setting where ROCO is used to pre-generate a pool of adversarial instances, which are then used together with clean data in our DPD framework. For each problem and size, we define a preference grid $\Lambda$ (Bi-objective: $|\Lambda|=101$; Tri-objective: $|\Lambda|=105$) and draw $M$ clean instances per $\lambda \in \Lambda$ from the uniform distribution used in the original solvers. Running ROCO on these clean instances under WS scalarization produces an adversarial set for each $\lambda$; pooling them yields $\mathcal{D}_{\text{hard}}^{\text{ROCO}}$. To ensure fairness, the per-$\lambda$ ROCO budget (number of adversarial instances or wall-clock time) is matched across $\lambda$ and aligned with the budget used in our PAA experiments.

**Training.** In each epoch, we build the training set

$$\mathcal{D}_{\text{train}} = \mathcal{D}_{\text{clean}} \cup \mathcal{D}_{\text{hard}}^{\text{ROCO}}.$$

Mini-batches are sampled by stratified sampling over $\lambda$ and data source (clean vs. ROCO), with a default 1:1 ratio. We keep the DPD pipeline unchanged: for each mini-batch we generate $N$ perturbed preferences $\{\lambda_i'\}_{i=1}^N$ in an $\epsilon$-neighborhood of the batch preference and renormalize them to the simplex; we compute Tchebycheff values for $\{\lambda_i'\}$, form relevance scores via Eq. (8), pick the weakest preference $\lambda_{\text{adv}}'$ by Eq. (9), and update the policy by REINFORCE using Eq. (11). All other optimization hyperparameters (optimizer, learning rate, batch size, RL baselines) follow the corresponding original solvers.

**Evaluation protocol.**    We evaluate on Gaussian-mixture test sets (200 instances per setting) with cluster spread $c_{\text{DIST}} \in \{10, 20, 30, 40, 50\}$ for Bi-TSP and Bi-CVRP at node sizes $\{20, 50, 100\}$. Metrics include mean HV (higher is better), mean relative HV gap (lower is better) computed against WS-LKH (MOTSP/MOCVRP).

**Results.**    As shown in Table 3, ROCO-trained models (our solvers trained with $\mathcal{D}_{\text{hard}}^{\text{ROCO}}$ under DPD) consistently improve HV and reduce optimality gaps over counterparts trained without DPD, while incurring negligible runtime overhead, validating that offline ROCO-adversarial data, when fed through our DPD scheme, yields robust gains under distribution shift.

Table 3: Defense performance when training with ROCO-adversarial instances.

| Method | Defense | Bi-TSP (50 nodes) | | | Bi-CVRP (50 nodes) | | |
|---|---|---|---|---|---|---|---|
| | | HV ↑ | Gap ↓ | Time ↓ | HV ↑ | Gap ↓ | Time ↓ |
| PMOCO | None | 0.8566 | 1.08% | 11.33s | 0.5219 | 1.79% | 18.02s |
| PMOCO | ROCO-DPD | 0.8583 | 0.88% | 13.41s | 0.5271 | 0.81% | 18.11s |
| PMOCO | PAA-DPD | 0.8654 | 0.06% | 13.29s | 0.5308 | 0.11% | 17.44s |
| CNH | None | 0.8587 | 0.84% | 12.03s | 0.5289 | 0.47% | 19.55s |
| CNH | ROCO-DPD | 0.8632 | 0.32% | 16.04s | 0.5345 | −0.58% | 19.23s |
| CNH | PAA-DPD | 0.8660 | 0.0% | 15.22s | 0.5386 | -1.35% | 19.98s |

# E    BENCHMARK EVALUATIONS

Similarly to previous studies Li et al. (2021a) Fan et al. (2024), we evaluated the performance of our DPD framework on six Bi-TSP100 benchmark instances[4]: kroAB100, kroAC100, kroAD100, kroBC100, kroBD100 and kroCD100, which were constructed by combining instances from the kroA100, kroB100, kroC100, and kroD100 instances.

As illustrated in Figure 5, models trained on the hard instances consistently outperform those trained on the clean instances in all the problem instances. The CNH-DPD model achieves the HV values among the learned models, closely approaching the exact PF. In particular, in kroAC100, PMOCO-DPD and CNH-DPD achieve HV values that are 1.4% and 1.7% higher than those of PMOCO and CNH, respectively. In kroBC100 and kroBD100, the HV values for DPD-enhanced models are within 0.1% of the exact PF, demonstrating their competitive performance and robustness. These results underscore the effectiveness of the proposed approach in handling diverse instance distributions and enhancing solver adaptability under adversarial conditions.

# F    GENERALIZATION STUDY

We evaluate the generalization capability of DPD on two types of larger scale test instances ($n = 150/200$) including clean instances and mixed Gaussian instances. As illustrated in Table 4, our model demonstrates remarkable robustness across both test scenarios while maintaining strong performance under varying instance distributions.

Table 4: Comparison of Bi-TSP performance with $n = 150$ and $n = 200$ on 200 clean and Mix Gaussian test instances.

| Method | Clean Instances | | | | | | Gaussian Instances | | | | | |
|---|---|---|---|---|---|---|---|---|---|---|---|---|
| | Bi-TSP ($n = 150$) | | | Bi-TSP ($n = 200$) | | | Bi-TSP ($n = 150$) | | | Bi-TSP ($n = 200$) | | |
| | HV | Gap | Time | HV | Gap | Time | HV | Gap | Time | HV | Gap | Time |
| WS-LKH | **0.7149** | - | 13h | **0.7490** | - | 22h | **0.8506** | - | 13h | **0.8790** | - | 22h |
| PMOCO | 0.7028 | 1.69% | 55.38s | 0.7318 | 2.29% | 1.52m | 0.8367 | 1.63% | 55.87s | 0.8608 | 2.07% | 1.52m |
| **PMOCO-DPD** | 0.7091 | 0.81% | 57.22s | 0.7327 | 2.17% | 1.59m | 0.8430 | 0.89% | 57.22s | 0.8660 | 1.47% | 1.59m |
| CNH | 0.7043 | 1.48% | 57.45s | 0.7324 | 2.21% | 1.53m | 0.8379 | 1.49% | 57.49s | 0.8598 | 2.18% | 1.53m |
| **CNH-DPD** | 0.7104 | 0.63% | 58.33s | 0.7374 | 1.54% | 2.02m | 0.8427 | 0.92% | 58.36s | 0.8649 | 1.60% | 2.05m |

---

[4]https://sites.google.com/site/kflorios/motsp?pli=1

