# OpenReview forum: "Towards Robust Multi-Objective Optimization: Adversarial Attack and Defense Methods for Neural Solvers"
_ICLR.cc/2026/Conference — ICLR 2026 Conference Withdrawn Submission_

### Official Review · Reviewer_3jZJ · 2025-10-25

**Soundness:** 2
**Presentation:** 3
**Contribution:** 2
**Rating:** 2
**Confidence:** 5

**Summary:**

This paper investigates the robustness of DRL-based solvers for multi-objective combinatorial optimization problems (MOCOPs). The authors propose PAA (Preference-based Adversarial Attack), which generates hard instances by performing gradient ascent on the RL loss for specific preference vectors, and DPD (Dynamic Preference-augmented Defense), which combines adversarial training with hardness-aware preference selection. Experiments on MOTSP, MOCVRP, and MOKP show that existing neural solvers are vulnerable to distribution shifts and that the proposed defense can improve performance on out-of-distribution instances.

**Strengths:**

1. **Important problem**: Even the methodology is not new, this appears to be the first systematic study of adversarial robustness for MOCOPs. Given the increasing deployment of neural solvers, understanding their failure modes is crucial.

2. **Comprehensive evaluation**: The paper tests on multiple problems (TSP, CVRP, KP) with different objective counts and evaluates against several state-of-the-art neural solvers (PMOCO, CNH, EMNH).

**Weaknesses:**

**1. Theoretical Foundation Issues**

The core hypothesis (Section 4.1) that
> if a neural model does not effectively approximate the solution of the subproblem under certain values of $\lambda$, the resulting approximation of the Pareto front will be inadequate

lacks both theoretical justification and empirical validation. The paper doesn't establish the mapping between preference vectors and Pareto front regions. Similarly, the PAA method lacks convergence analysis (on theory or experiment), otherwise, why attack the RL loss rather than solution quality directly?

**2. Serious Implementation Inconsistencies**

The paper has several technical inconsistencies that undermine its credibility:
- Algorithm 1 line 6 has $D_{hard} = D_{hard} \cup d^{adv,k}$ which causes infinite dataset growth
- PAA uses division for baseline (Eq. 4) while DPD uses subtraction (Eq. 11) without explanation
- Baseline solvers use weighted sum but DPD training uses Tchebycheff decomposition
- The preference perturbation in DPD is logically flawed - hard instances are generated for $\lambda$ but then trained with $\lambda^{'}_{adv}$, breaking the attack-defense correspondence

**3. Unfair Experimental Setup**

The comparison is fundamentally unfair in multiple ways:
- DPD uses ~2x training data (10,000 clean + 10,100 hard instances vs 10,000 for baseline)
- DPD gets 200 epochs of fine-tuning on a pre-trained model while baselines don't
- No ablation studies to separate the effects of extra data, extra training, and the actual method

**4. Insufficient Evaluation**

The evaluation relies solely on HV and gap metrics, which can be dominated by outliers. Missing standard metrics like IGD (convergence) and Spread (diversity). Tables only report relative gaps without objective values. Most critically, there's no ablation study to verify whether improvements come from the proposed method or simply from having more training data/time.

**Questions:**

1. **On the core hypothesis**: Can you provide empirical evidence showing how attacking specific λ values affects corresponding regions of the Pareto front? How do you handle the fact that multiple λ values can map to the same Pareto solution?

2. **On fairness**: What happens if you give the baseline models the same amount of training data and fine-tuning epochs? Can you show results where total training samples are controlled?

3. **On preference perturbation**: In Algorithm 1, you generate hard instances for λ but then train with perturbed λ'. Isn't this a mismatch? Shouldn't you generate new instances for λ' or use the original λ for training?

4. **On convergence**: Can you show that the PAA attack actually converges? What's the stopping criterion beyond "converges towards hard instances"?

5. **On decomposition**: Why use WS-based baselines when your defense uses Tchebycheff? This seems to favor your method. Are there Tchebycheff-based exact solvers available?

6. **On computational cost**: What's the actual wall-clock time for generating hard instances and fine-tuning? The paper only reports inference time which is misleading about the true computational cost.

7. **On ablation**: Can you separate the contributions of: (a) PAA-generated instances, (b) preference perturbation, (c) hardness-aware selection, and (d) simply having more training data?

8. **On Algorithm 1**: Is line 6 a typo? Should $D_hard$ be reset each epoch rather than accumulated?

9. **On training from scratch**: Can you show the performance of DPD when training neural solvers from scratch rather than fine-tuning? I'm curious whether the method still works effectively without relying on pre-trained models, as this would better demonstrate the generality of the approach.

---

### Official Review · Reviewer_9SUY · 2025-10-26

**Soundness:** 2
**Presentation:** 3
**Contribution:** 2
**Rating:** 2
**Confidence:** 3

**Summary:**

This work proposes an adversarial attack on reinforcement learning based neural combinatorial for multi-objective combinatorial problems, as well as an adversarial defense derived from it.

The main insight is that RL-based solvers can be trained for multi-objective optimization by mapping vectors of objective values $(f_1(\pi, s), f_2(\pi, s), \dots, f_N(\pi, s))$ for solutions $\pi$ and problem instanceses $s$  to a scalar value based on preference vectors from $\Lambda \subseteq \mathbb{R}^N$ (e.g., via weighted sums). Adversarial instances can thus be constructed via (projected) gradient-based optimization of these scalar "subproblems" for various $\lambda \in \Lambda$.

As an adversarial defense, the authors propose to not only perform adversarial training on the thusly generated hard problem instances, but to simultaneously augment the used set of preference vectors $\Lambda$ in an adversarial manner.

In the experimental evaluation, model performance is not evaluated on scalar "subproblems", but standard metrics that capture the Pareto-optimality of the produced solutions to the multi-objective problems.
On clean instances sampled from simple distributions (e.g., uniform, log-normal) the proposed adversarial attack compares favorably to prior work and randomly sampled perturbations.
The proposed adversarial defense is subsequently shown to improve generalization from clean, uniformly sampled problem instances to clean, Gaussian problem instances.

**Strengths:**

* The attack objective is both natural and nicely motivated by references to prior work on non-adversarial multi-objective optimization
* Adversarially augmenting the preference vectors as well makes for a somewhat more interesting method than conducting standard adversarial training
* The experiments cover a sufficiently wide range of models, problem types (traveling salesman, knapsack etc.), and problem instances
* The manuscript is generally well-written and easy to follow
* Studying multi-objective optimization from an adversarial perspective appears to be novel and could thus be of interest to both the neural combinatorial optimization community and the trustworthy ML community

**Weaknesses:**

* While interesting, the proposed defense is not sufficiently motivated. Assuming that it is purely empirically motivated, then the gain from using adversarial preferences should be demonstrated by the authors via an ablation study. I would suggest considering a Cartesian product of the following, or similar: (Clean training instances, randomly perturbed traning instances, adversarial training instances) x (Clean preferences, randomly perturbed preferences, adversarially chosen preferences)
* Similarly, the evaluation of the proposed defense is not sufficiently motivated. Training on uniform instance and evaluating on Gaussian instances seems to suggest to me that the defense is ultimately meant as a general tool for improving generalization performance. However, the abstract explicitly states that the goal was to "mitigate the adversarial effect" (l.20), which is not captured by the experiments.
* The HV metric is not sufficiently explained and motivated. For the benefit of future readers, I would suggest including an explanation & discussion in the appendix, instead of just pointing to another paper.
* In computing the reported metrics, non-neural solvers ("WS-LKH" and "WS-DP") are used. The authors do not sufficiently discuss (a) whether these solvers are exact and (b), if not, how their potential non-robustness to attacks may influence the results.
* No code is provided, so reproducibility is poor

### Minor comments
* In Section 3.2 "$g_{(\cdot)}$" is used for scalar subproblems, whereas later $L(\cdot)$ is used
* In reinforcement learning literature $\pi$ is often used to denote the policy. Using $\pi$ for the solution (even though it is natural for TSP) is a little confusing
* The "baselines" $b$ and $L_b$ in Eq. 4 and Eq. 11 are not sufficiently explained

**Questions:**

* Did I understand you correctly, that Table 2 does not show adversarial generalization, but generalization to a completely different data distribution?
* If it is supposed to show adversarial generalization: Do you evaluate on the $\mathrm{D}_\mathrm{hard}$ from Algorithm 1, or do you compute new adversarial examples after Algorithm 1 has been applied?

---

### Official Review · Reviewer_gPtn · 2025-10-31

**Soundness:** 3
**Presentation:** 3
**Contribution:** 1
**Rating:** 2
**Confidence:** 4

**Summary:**

This paper proposes to borrow the idea of adversarial attack and defense to enhance neural solvers for multi-objective optimization. Specifically, it uses adversarial samples as hard instances and involve them during training. Experimental results demonstrate that the adversarial samples can trap the solvers, and involving adversarial samples during training can facilitate the solvers to achieve better performance.

**Strengths:**

1. The paper-writing is generally clear.
2. The experimental results demonstrate that associated with various neural solvers, the introduced method can consistently achieve improvements.

**Weaknesses:**

The key idea of this paper, generating adversarial samples via gradient and utilizing them during training, is prevailing. Adapting it in neural solvers for multi-objective optimization can bring expected improvements, but it remains mostly incremental combination. Therefore, I am concerned that the novelty and contribution of this work can hardly reach the bar of ICLR.

**Questions:**

In Table 1, it seems that for the permutation methods of LogNormal(0, 1), Beta(2, 5) and Gamma(2, 0.5), the solvers can even consistently achieve better optimality gaps on randomly permutated instances (except for Bi-KP). Are there any explanations or discussions on it?

---

### Official Review · Reviewer_DDkq · 2025-11-04

**Soundness:** 2
**Presentation:** 3
**Contribution:** 3
**Rating:** 4
**Confidence:** 4

**Summary:**

This paper presents an adversarial attack and defense framework for RL solvers for multi-objective combinatorial optimization. The authors proposed a white-box attack method and evaluated several solvers trained with RL. Different attach methods, including random attack and ROCO (ICLR'23), are compared in bi-objective TSP, tri-objective TSP, bi-objective CVRP, and bi-objective KP.

**Strengths:**

* This paper presents a general benchmarking framework and evaluates the robustness of RL-based multi-objective CO solvers.
* The experimental evaluation is rigorous and extensive, and the authors highlight the strength of the proposed attack method and the adversarial training defense strategy.

**Weaknesses:**

* In the proposed attacking framework, there seems to be no guarantee that the generated adversarial examples are directly comparable to the original ones in terms of HV and Gap. In contrast, the compared adversarial attack method, ROCO, guarantees that the new optimal objective is no worse than that of the original problem.

  I believe this is an important aspect to consider because one can trivially generate adversarial samples with a 100% gap: for a TSP problem, just make all distances on the optimal route to be 0, and keep the others to be non-zero. Any non-optimal solution now has a gap of 100%. These are trivial examples, but they seem to be considered as "successful attacks" in the context of this paper. They are also easy to overfit by the model if fed into the adversarial training loop, but those examples do not contain much knowledge to learn.

* Table 1 is a bit overwhelming and hard to understand. Please try to explain the rows and columns in the title.
* The technical contribution of this manuscript is a bit incremental compared to previous adversarial attack and defense works on single-objective optimization problems.

**Questions:**

* Are there any theoretical guarantees of the adversarial examples? Is it possible to incorporate such constraints as ROCO did? I will be more convinced about the soundness of this manuscript if the authors can describe the Gap and HV of their generated instances are directly comparable with the ones from the original instance, or they can provide

---

### Note · Authors · 2025-12-01

I have read and agree with the venue's withdrawal policy on behalf of myself and my co-authors.